# Case Series Evaluating the Relationship of SGLT2 Inhibition to Pulmonary Artery Pressure and Non-Invasive Cardiopulmonary Parameters in HFpEF/HFmrEF Patients—A Pilot Study

**DOI:** 10.3390/s25030605

**Published:** 2025-01-21

**Authors:** Ester Judith Herrmann, Michael Guckert, Dimitri Gruen, Till Keller, Khodr Tello, Werner Seeger, Samuel Sossalla, Birgit Assmus

**Affiliations:** 1Department of Medicine I, Cardiology, University Hospital Giessen and Marburg, 35392 Giessen, Germany; ester.herrmann@innere.med.uni-giessen.de (E.J.H.); samuel.sossalla@innere.med.uni-giessen.de (S.S.); 2Institute of Mathematics, Natural Sciences and Data Processing, Technische Hochschule Mittelhessen—University of Applied Sciences, 61169 Friedberg, Germany; michael.guckert@kite.thm.de; 3Cognitive Information Systems Group, Kompetenzzentrum für Informationstechnologie (KITE), Technische Hochschule Mittelhessen—University of Applied Sciences, 61169 Friedberg, Germany; 4Department of Internal Medicine I, Cardiology, Justus-Liebig-University Giessen, 35392 Giessen, Germany; d.gruen@kerckhoff-fgi.de (D.G.); till.keller@med.uni-giessen.de (T.K.); 5German Center for Cardiovascular Research (DZHK), Partner Site Rhein Main, 61231 Bad Nauheim, Germany; 6Department of Medicine II, Internal Medicine, Pneumology, University Hospital Giessen and Marburg, 35392 Giessen, Germany; khodr.tello@innere.med.uni-giessen.de (K.T.); werner.seeger@innere.med.uni-giessen.de (W.S.)

**Keywords:** heart failure, SGLT2 inhibitors, pulmonary artery pressure, RV-PA coupling, pulmonary vascular resistance

## Abstract

The initiation of sodium–glucose cotransporter 2 (SGLT2) inhibitor treatment was shown to reduce pulmonary artery pressure (PAP) in New York Heart Association (NYHA) class III heart failure (HF) patients with an implanted PAP sensor. We aimed to investigate the impact of SGLT2-I initiation on pulmonary vascular resistance (PVR), pulmonary capillary wedge pressure (PCWP), pulmonary arterial capacitance (PAC), and right ventricle (RV) to PA (RV-PA) coupling in a pilot cohort of HF with preserved/mildly reduced ejection fraction (HFpEF/HFmrEF) patients and whether PVR and PCWP can be serially calculated non-invasively using PAP sensor data during follow-up. Methods: Right heart catheterization parameters (PVR, PCWP, and PAC) were obtained at sensor implantation and echocardiographic assessments (E/E’, RV-PA coupling, and RV cardiac output) were made at baseline and every 3 months. SGLT2 inhibition was initiated after 3 months of telemedical care. Three methods for calculating PVR and PCWP were compared using Bland–Altman plots and Spearman’s correlation. Results: In 13 HF patients (mean age 77 ± 4 years), there were no significant changes in PAP, PVR, PCWP, RV-PA coupling, or PAC over 9 months (all *p*-values > 0.05), including after SGLT2-I initiation. PVR values were closely correlated across the three methods (PVR_New_ and PVR_New Tedford_ (r = 0.614, *p* < 0.001), PVR_Echo_ and PVR_New Tedford_ (r = 0.446, *p* = 0.006), and PVR_Echo_ and PVR_New_ (r = 0.394, *p* = 0.016)), but PCWP methods lacked reliable association (PCWP_Echo_ and PCWP_New_ (r = 0.180, *p* = 0.332). Conclusions: No changes in cardiopulmonary hemodynamics were detected after hemodynamic telemonitoring either prior to or following SGLT2-I initiation. Different PVR assessment methods yielded comparable results, whereas PCWP methods were not associated with each other. Further investigations with larger cohorts including repeated right heart catheterization are planned.

## 1. Introduction

Sodium–glucose cotransporter 2 (SGLT2) inhibitors (SGLT2-Is) are considered to be the new “miracle weapon” in the treatment of heart failure (HF) with preserved and mildly reduced ejection fraction (HFpEF and HFmrEF), with a class I recommendation in the revised European Society of Cardiology (ESC) 2023 guidelines for the treatment of acute and chronic HF [1]. SGLT2-Is have pleiotropic effects, including the improvement of glycemic control by blocking glucose reabsorption in the proximal convoluted tubule of the kidney and thereby reducing oxidative stress in the kidneys [2], and they were shown to reduce HF patient hospitalizations and mortality [3,4] and to increase exercise capacity and quality of life [5,6]. The cardiovascular benefits manifest rapidly and are unlikely to be related to the described primary mechanism of improved glycemic control. Further effects include the enhancement of early natriuresis, reductions in plasma volume, improved vascular resistance, reduced blood pressure, and changes in tissue sodium handling and might underlie the rapid reduction in the risk of HF [2]. Nevertheless, the mechanisms of action of SGLT2-Is on the cardiovascular system are thus far incompletely understood.

In patients with HF, even mild elevation in pulmonary vascular resistance (PVR) is associated with adverse outcomes [7]. It is not yet known whether SGLT2-Is can also diminish PVR and pulmonary capillary wedge pressure (PCWP) or increase pulmonary arterial capacitance (PAC) or right ventricle to pulmonary artery (RV-PA) coupling and, thereby, have a beneficial effect in HFpEF/HFmrEF patients with different extents of pre- and postcapillary pulmonary hypertension (PH). The echocardiographic estimation of PVR is limited in approximately 50% of HF patients [8] because it requires a pulmonary regurgitation signal. However, a combination of PAP sensor-derived pulse pressure (PP= PAPsystolic—PAPdiastolic) and echocardiography may provide superior results. Similarly, E/E’ was shown to correlate with PCWP with an acceptable accuracy at rest in cases with an elevated PCWP [9], but E/E’ is sometimes challenging to measure. Thus, a combination of echocardiography and (non)-invasive measurements derived from a PAP sensor could be helpful.

Therefore, the aim of the present pilot trial of HFmrEF and HFpEF patients was to investigate the potential of de novo SGLT2 inhibition to interact with cardiopulmonary parameters, with a focus on changes in PVR, PCWP RV-PA coupling, and pulmonary artery compliance. Specifically, we aimed to compare different methods for estimating PVR and PCWP by combining echocardiography and (non)-invasive examinations.

## 2. Materials and Methods

Patients with chronic HF with preserved and mildly reduced ejection fraction (HFpEF/HFmrEF, LVEF ≥ 45%) in New York Heart Association (NYHA) functional class III and with a cardiac decompensation event within the last 12 months were offered implantation of a PAP sensor (CardioMEMS^TM^, Abbott, Sylmar, CA, USA) and participation in a multi-center telemonitoring registry. Patients received PAP-guided HF management between 2020 and 2023 and were repeatedly trained in HF self-care by an ESC-certified HF nurse. We excluded participants who were not able to reassemble medication changes by telephone contact and who had a glomerular filtration rate of less than 30 mL/min.

### 2.1. Guideline-Directed Medical Therapy (GDMT)

Optimal medical therapy had been initiated and up-titrated as individually tolerated, as recommended in the ESC Guidelines of 2021 [10]. After a baseline monitoring period of three months and after publication of the ESC Guidelines in 2023 [1], SGLT2-I therapy was initiated in our advanced, elderly HF cohort (Figure 1).

### 2.2. Pulmonary Vascular Resistance and Pulmonary Capillary Wedge Pressure

PVR and PCWP were measured invasively via right heart catheterization (RHC) at baseline. Follow-up assessments were carried out using the following different non-invasive sensor-derived and/or echocardiography-derived approaches every 3 months.

The resistance/capacitance method:

It has been shown that the product of PVR and PAC (with PAC calculated as stroke volume (SV)/pulse pressure (PP = PAs-PAd); also denoted as PA compliance), the RC time (RC = PVR × PAC), is a given constant for each individual patient. Both in healthy individuals and in patients with PH, the PVR and PAC are thus inversely related, with their product forming a patient-specific constant value even under conditions of major changes such as response to therapy. This patient-specific RC constant is obtained from the baseline RHC measurements. Changes in PP are followed by telemonitoring (CardioMEMS^TM^) and can directly be converted into changes in PVR to obtain the current PVR (PVR_New_) at any time of the follow-up investigation, as long as the current SV is known. Regular follow-up echocardiography measurements for estimation of the current SV (SV_New_; based on the RV outflow tract diameter and velocity time integral) were thus performed, allowing the calculation of the current PAC in combination with the respective telemonitoring-based PP, and—based on the known RC time of the individual patient—the current PVR (PVR_new_):PVR_New_ = RC time × PP_New_/SV_New_(1)

Moreover, this approach allows the non-invasive calculation of the PCWP at this point of echocardiographic assessment: the diastolic pressure gradient (DPG) of the last RHC measurement (DPG_RHC_) is corrected for the current SV as compared to the SV of the last RHC (SV_new_ and SV_RHC_, respectively) and the current PVR as compared to the PVR of the last RHC (PVR_New_ and PVR_RHC_, respectively) to give the current DPG (DPG_New_):DPG_New_ = DPG_RHC_ × PVR_new_/PVR_RHC_ × SV_New_/SV_RHC_(2)

The current PCWP (PCWP_new_) is then calculated asPCWP_New_ = PAPd_CardioMEMS_ − DPG_New_(3)

Thus, the combination of precise measurement of PAP by hemodynamic telemonitoring and the intermittent assessment of the SV by echocardiography should allow non-invasive follow-up of cardiac output (CO), PVR, and PCWP, key features of HFpEF/HFmrEF patients with and without PH in addition to the direct PAP tracing.

2.The resistance/capacitance method corrected by Tedford:

With respect to the patient-individual RC constant, it has been discussed whether, in patients with elevated PCWP, the RC time shows a decline with increasing PCWP [11]. Thus, in an additional approach, we adopted a formula suggested by Tedford and colleagues [12] for the correction of RC under conditions of elevated PCWP values (RC =−0.0063 × PCWP + 0.46) with a PCWP >15 mmHg, with further calculations adapted accordingly.

3.Echocardiographic-based assessment:

PVR_Echo_ (in WU) was measured as the ratio of peak tricuspid regurgitant velocity (TRV, in ms) to the RV outflow tract time–velocity integral (TVI_RVOT_, in cm) using the following formula [13]:PVR_Echo_ = TRV/TVI_RVOT_ × 10 + 0.16 (4)

PCWP_Echo_ was assessed by measuring E/E’_baseline_, where E/E’ is the ratio of the peak mitral inflow (*E*-wave) velocity to the early diastolic tissue Doppler mitral annular velocity (E’). E/E’ appears to have a modest correlation with PCWP and an acceptable accuracy for elevated PCWP at rest [9]. The E/E’_baseline_ was then equilibrated with the RHC-based PCWP and subsequently measured every 3 months by echocardiography. PCWP_Echo_ was then calculated by the following formula:PCWP_Echo_ =E/E’ _Follow-up_ × PCWP_RHC_/E/E’_baseline_(5)

### 2.3. Pulmonary Arterial Capacitance

The PAC or compliance was measured as ratio of the echocardiography-derived RV SV and the non-invasively assessed PP from the implanted PA sensor.PAC_New_ = SV_New_/PP_New_(6)

In addition, core laboratory NT-proBNP levels and hematocrit were assessed every three months.

All patients who received the PAP sensor (CardioMEMS^TM^) provided written informed consent for participation in the ongoing registry (NCT03020043, https://clinicaltrials.gov/study/NCT03020043?tab=history&a=2, accessed on 13 January 2017). This study was approved by the local ethics committee and complied with the principles laid out in the Declaration of Helsinki.

### 2.4. Statistical Analysis

The present case series is a descriptive analysis of one cohort over 9 months with an implanted PAP sensor. Baseline characteristics were calculated as the median and interquartile range (IQR) or mean and standard deviation (SD), as appropriate, after testing for normal distribution with the Shapiro–Wilk test. The mean PP was calculated as the difference between PAPsystolic and PAPdiastolic and proportional PP as the ratio of PP to PAPsystolic. Pairwise comparisons were performed between baseline and, with empagliflozin initiation at 3 months, to 6- and 9-month follow-up results, assessing PAP values and laboratory and echocardiographic parameters with the non-parametric Friedman test or ANOVA for repeated measurements, where appropriate. Comparisons of the different formulas for measuring PVR and PCWP were depicted in Bland–Altman plots and the correlations between the different formula were calculated by the non-parametric Spearman’s correlation. Values were only integrated in the Bland–Altman plots if available from both techniques. Mean PAP values for all patients 3 months prior to and over the duration of 6 months after the initiation of SGLT2-I were calculated and illustrated with LOWESS regression.

Statistical significance was assumed if *p* < 0.05 and all reported *p*-values are two-sided and with Bonferroni correction for repeated measurements. Statistical analysis was carried out with the software packages SPSS (Version 29.0.1.1, SPSS Inc., IBM Corp., Armonk, NY, USA) and in Python (Version 3.9, Python Software Foundation, Wilmington, Delaware, USA) and the packages numpy (Version 2.1.2), pandas (Version 2.2.3), matplotlib (Version 3.9.2), and statsmodels (Version 0.14.4).

## 3. Results

### 3.1. Baseline Characteristics

The cohort comprised a total of 13 chronic HF patients with functional NYHA class III. The mean age was 77 ± 4 years, 61.5% were female, and 84.6% were diagnosed with HFpEF and 15.4% with HFmrEF. A total of 23.1% had diabetes mellitus type 2, 69.2% atrial fibrillation, and 92.3% arterial hypertension (Table 1).

The median LVEF was 60 (53–60)% and the mean cardiac index 1.9 ± 0.3. The mean PAP values at the time of right heart catheterization immediately prior to PAP sensor implantation were as follows: systolic PAP 46 ± 13 mmHg, mean PAP 28 ± 9 mmHg, and diastolic PAP 16 ± 7 mmHg. The mean PCWP was 20 ± 6 mmHg, suggestive of persistent congestion, the mean pulse pressure (calculated as systolic PAP minus diastolic PAP) was 29 ± 11 mmHg, and the mean proportional pulse pressure was 0.64 ± 0.14. The median pulmonary artery compliance (PAC) was 1.7 (interquartile range (IQR): 1.3–2.6) mL/min, and the median PVR was 2.1 (IQR: 1.6–4.3) Wood units. Eleven of thirteen patients (84.6%) had PH (PAPmean > 20 mmHg) and, of those, four (30.8%) presented with a combined pre-and post-capillary PH phenotype, with a PVR > 2 WU, according to the ESC 2022 Guidelines [14].

### 3.2. Temporal Course of Hemodynamic and Laboratory Parameters Prior to and After SGLT2 Inhibitor Therapy

During the first 3 months of PAP-guided HF therapy prior to SGLT2 initiation, as well as 6 months after initiation of SGLT2-I therapy, hemodynamic monitoring of diastolic, mean, and systolic PAP revealed stable pressures and no significant change in any value (3 months prior to SGLT2-I: PAP diastolic: 22 ± 7 mmHg; PAP mean: 32 ± 9 mmHg; PAP systolic 46 ± 12 mmHg, 6 months post-SGLT2-I: PAP diastolic: 21 ± 5 mmHg, p^3^ = 1.000; PAP mean: 31 ± 7 mmHg, p^3^ = 1.000; PAP systolic 45 ± 9 mmHg, p^3^ = 1.000, Table 2).

Likewise, at 6 months, neither the hematocrit (3 months prior to SGLT2-I: 0.39 ± 0.02%; 6 months post-SGLT2-I: 0.40 ± 0.04%, p^3^ = 1.000) nor the NT-proBNP levels (3 months prior to SGLT2-I: 1412 (562–2011) pg/mL; 6 months post-SGLT2-I: 1017 (568–3513) pg/mL, p^3^ = 0.375) changed significantly (Table 3).

It is of note that, during the entire follow-up, a total of 36 adjustments of GDMT and 47 adjustments of diuretic therapy were performed under the guidance of hemodynamic monitoring (Figure 2). In detail, within the first 3 months prior to the initiation of SGLT2-I, we changed the dosage of diuretics in our patient cohort 19 times and the GDMT dosage 13 times. Subsequently, in the first 3 months after the start of the SGLT2-I therapy, there were 16 changes of diuretics and 18 changes of GDMT and in the next 3 months up to 6 months after the start of the SGLT2-I therapy there were 12 changes of diuretics and 5 changes of GDMT (Figure 2).

Similar to the case with PAP, neither PVR nor PCWP showed significant changes using the three different approaches (see Methods) during 9 months of follow-up (Table 2). A significant association was observed between all three different techniques of estimating PVR non-invasively, with the strongest association between PVR_New_ and PVR_New Tedford_ (r = 0.614, *p* < 0.001), whereas there were weaker associations between PVR_Echo_ and PVR_New Tedford_ (r = 0.446, *p* = 0.006) and PVR_Echo_ and PVR_New_ (r = 0.394, *p* = 0.016) (Figure 3A–C). No association was found for the non-invasive estimation of PCWP using PCWP_Echo_ and PCWP_New_ (r = 0.180, *p* = 0.332; Figure 3D) according to the formulas provided in the Methods Section.

Individual comparisons of the three different techniques are shown as Bland–Altman plots in Figure 4.

We further assessed PAC, which was shown to be a prognostic marker, especially in HFpEF patients. However, no change in PAC was found during the 9 months of follow-up and after initiation of SGLT2-I therapy (3 months prior to SGLT2-I: 3.5 (3.1–5.2) mL/mmHg; 6 months after SGLT2-I: 3.3 (2.2–4.6)] mL/mmHg, p^3^ = 1.000) (Table 2).

Finally, RV-PA coupling, which reflects the matching of RV contractility to RV afterload, was analyzed during 9 months of follow-up. Our data show that RV-PA coupling was moderately impaired throughout the study course, and no change in RV-PA coupling was observed during the 9 months of follow-up and after initiation of SGLT2-I therapy (3 months prior to SGLT2-I: 0.48 ± 0.15 mm/mmHg; 6 months after SGLT2-I: 0.48 ± 0.13 mm/mmHg, p^3^ = 1.000) (Table 2).

Similar results for all variables were observed after 3 months of SGLT2-I therapy (Table 2).

## 4. Discussion

The current prospective case series is, to the best of our knowledge, the first investigation of the impact of an SGLT2-I on non-invasively assessed PVR, PCWP, RV-PA coupling, and PAC in an HFpEF/HFmrEF patient cohort managed by PAP-guided HF care. We did not detect any significant change in any of these parameters in the short-term follow-up of 6 months after SGLT2-I initiation. The findings of our case series reveal several new aspects in the context of HFpEF and HFmrEF: although we were able to confirm the general feasibility of serial non-invasive calculation of PVR and PCWP using values derived from implanted PAP sensors in combination with echocardiographic data, these tools demonstrated only a close association of the different formulas for the serial calculation of PVR but not for PCWP.

A possible explanation for this lack of association of the different techniques of estimating PCWP at rest may be that the calculation of PCWP_New_ by the resistance/capacitance method relies on the echocardiography-derived SV and PCWP_New Echo_ estimates PCWP by the echocardiography-based approach using E/E’. Both values are specifically prone to variation in patients with atrial fibrillation (69.2% in our study), although we calculated the mean of several cardiac cycles. In addition, it has been shown previously that invasively assessed PCWP and E/E’ [15] appear to have just a modest correlation and a worse accuracy in cases with elevated PCWP [9]. Of note, our derived PCWP_Echo_ formula even used the E/E’ values twice (at baseline and at follow-up), and this approach might have potentiated possible errors or further diminished the just modest correlation of E/E’ and PCWP. However, the non-invasive reliable and serial estimation of PVR and PCWP remains of considerable interest for patients undergoing hemodynamic PAP sensor-based telemonitoring. For example, a rise in PAP could either reflect increasing PVR or increasing PCWP, which would be treated differently. Therefore, in a future study we will further address this question with a serial right heart catheter in parallel with the non-invasive estimations to validate the findings.

Interestingly, SGLT2-I therapy initiation did not lead to changes in cardiopulmonary interaction parameters such as RV-PA coupling or PAC, despite beneficial long-term effects on clinical outcome have been demonstrated within the last five years across the full spectrum of left ventricular ejection fraction. Therefore, one may assume that these drugs also positively interact with prognostic parameters of pulmonary vascular function. It is of note that, in HFpEF/HFmrEF patients, RV-PA uncoupling, which describes the insufficient contractility of the RV in response to an elevated afterload, is an independent predictor of adverse outcomes [16,17]. Whereas the PVR reflects the static component of RV afterload, the PAC (ratio of SV to pulmonary PP) represents the distensibility of the pulmonary artery tree, including both the static and dynamic pulsatile components [18,19]. TAPSE/PASP have been shown to have a linear correlation with PAC [20] and PAC provides an even a stronger prediction of RV failure and outcome than PVR and PCWP [19,21], especially in early phases of the disease where the PAC can be reduced but PVR is still normal [18]. In HFpEF/HFmrEF patients, impaired PAC is a sensitive early marker of the beginning of RV-PA uncoupling [16]. In our case series, the RV-PA coupling value of 0.48 ± 0.15 mm/mmHg was moderately reduced and the PAC of 3.5 (3.1–5.2) mL/mmHg was within the lower normal range and showed no changes during the entire follow-up.

The product of compliance (PAC) and resistance (PVR), the RC time, is a constant and the two factors have an inverse relationship. Increases in PCWP result in lower PAC but do not necessarily increase PVR; thus, the RC time decreases with increasing PCWP, and the exact position of the RC time is dependent on PCWP [22]. In patients with early-stage PH, an acute increase in left atrial pressure does not lead to an increase in PVR due to the recruiting of additional pulmonary vascular capacitance [23,24]. This compensatory mechanism is progressively lost as the severity of PH increases and, therefore, acute increases in the left atrial pressure can lead to an immediate increase in PVR, thus increasing the risk of RV decompensation and failure [25]. For example, in an advanced HFrEF cohort with 724 patients, with a PAC < 2.5 mL/mmHg and PVR > 2.1 WU, the odds ratio of having RV failure was 6.45 (4.35–9.67, *p* < 0.001) compared to patients with a high PAC and low PVR [22].

Nakagawa et al. have demonstrated a weak but concordant relationship between echocardiographic and RHC-assessed PAC in HFpEF patients who were hospitalized for acute decompensation [19]. Importantly, Al-Naamani et al. [21] reported that PAC was able to predict adverse outcomes more accurately than PVR in patients with HFpEF.

In our cohort, we had the advantage of direct access to daily invasive PAP measurements from implanted sensors, in conjunction with serial echocardiographic measures of SV for the calculation of PVR. This resulted in complete data acquisition, which is frequently not the case if the calculation of PVR relies on pure echocardiographic assessment, which might be limited by a lack of a pulmonary regurgitation signal in about 50% of HFpEF patients [8]. Our study emphasizes the potential for simplifying PVR monitoring in clinical practice, particularly in telemedicine-based heart failure care, and assessed an agreement between the different non-invasive techniques by using Bland–Altman plots and Spearmans’ correlation. However, the proposed use of the resistance/capacitance method as a combination of non-invasive hemodynamics-derived PP from a PAP sensor and echocardiographically measured SV in the case of PVR_New_, PVR_New Tedford_, PCWP_New_, and PAC_New_ requires validation in future invasive studies.

One finding of our case series deserves further discussion: the lack of a significant reduction in PAP, despite hemodynamically guided HF care and the initiation of SGLT2-I therapy. Our cohort consisted of 13 HFpEF/HFmrEF patients with a mean cardiac index of 1.9 ± 0.3, a mean PAP of 28 ± 9 mmHg, and a mean PCWP of 20 ± 6 mmHg prior to the start of treatment with an SGLT2-I. We did not observe any reduction in PAP or PVR and there was no increase in PAC during the follow-up, which might be associated with the fact that, on average, patients were still congested. This is suggested by a PCWP > 15 mmHg, estimated with the two different methods (PCWP_New_ 23 (21–27) and PCWP_echo_ 17 (16–25) mmHg), the still-elevated NT-proBNP (1017 (568–3513) pg/mL), and the reduced hematocrit (0.40 ± 0.04%) at the 9-month follow-up. On the other hand, in the MEMS-HF study, decreases in PAP were greater in patients with a baseline mean PAP ≥ 35 mmHg versus < 35 mmHg in the total MEMS-HF cohort [26], whereas in our cohort, the mean PAP was 32 mmHg at baseline. In our 13 patients, a total of 83 medication adjustments were performed during the entire 9-month follow-up time, suggesting that further lowering of PAP was not possible in this medically optimized HFpEF/HFmrEF cohort. However, the recently published randomized multicenter MONITOR-HF study from the Netherlands convincingly demonstrated that, independent of modern HF therapy and various baseline characteristics, PAP-guided HF treatment was superior to conventional treatment [27,28].

## 5. Limitations

Although this is a prospective analysis, the small patient number of the present case series is the major limitation of this study. Likewise, the missing effects of SGLT2-I on diastolic PAP, PVR, RV-PA coupling, and PAC could have been due to GDMT and diuretic adjustments that were implemented in cases with increasing PAP. Furthermore, interobserver variability in the echocardiographic assessment might have influenced the data, although echocardiographic data analysis was performed by staff who were unaware of patients’ medication and follow-up time in the study.

## 6. Conclusions

Using implanted PAP sensor data in combination with serial echocardiographic data allows for non-invasive follow-up assessment of PVR, but not for PCWP. In addition, within the present case series of 13 HFpEF and HFmrEF patients, we did not observe any change of cardiopulmonary interaction during 9 months of follow-up, despite hemodynamically guided HF care and SGLT2-I initiation. Further mechanistic hemodynamic studies are therefore planned to assess the potential impact of SGLT2 inhibition on cardiopulmonary interactions, which will be validated by invasive hemodynamics.

## Figures and Tables

**Figure 1 sensors-25-00605-f001:**
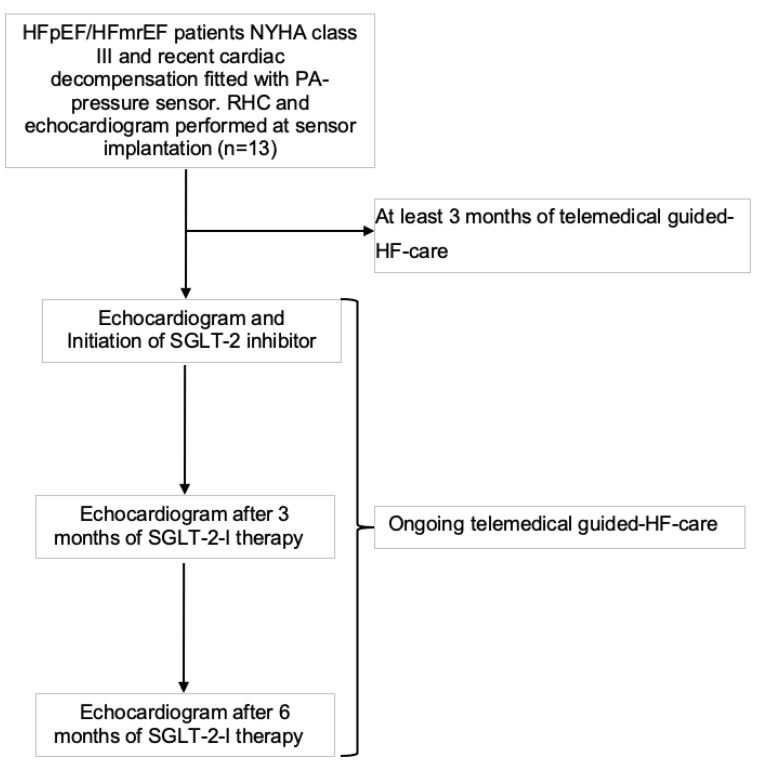
Study flowchart. HFpEF, heart failure with preserved ejection fraction; HFmrEF, heart failure with mildly reduced ejection fraction; RHC, right heart catheterization; SGLT-2-I, sodium–glucose cotransporter 2 inhibitor; PA, pulmonary artery; NYHA, New York Heart Association.

**Figure 2 sensors-25-00605-f002:**
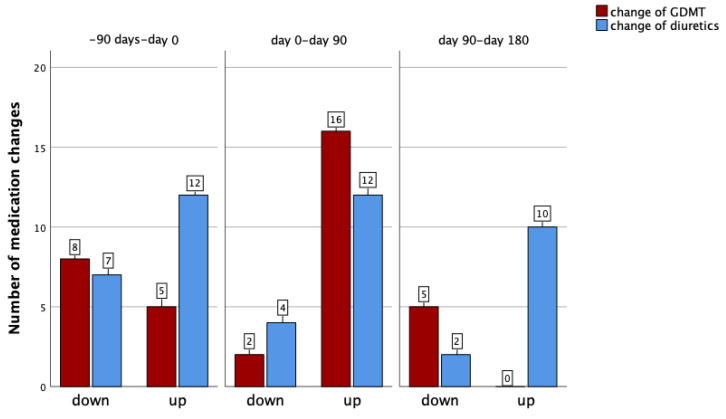
Medication changes for guideline-directed medical therapy (GDMT) or for diuretics during various study time periods.

**Figure 3 sensors-25-00605-f003:**
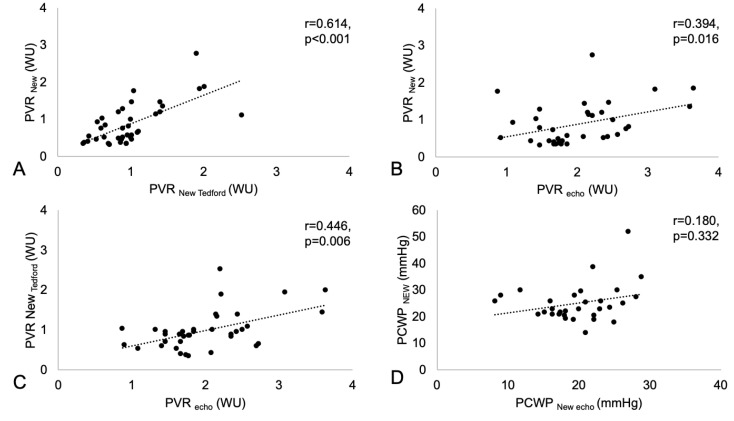
Association between pulmonary vascular resistance (PVR) values calculated by various methods. (**A**): PVR_New_ and PVR_New Tedford_, (**B**): PVR_New_ and PVR_Echo_, and (**C**): PVR_New Tedford_ and PVR_Echo_. (**D**): Association of pulmonary capillary wedge pressure (PCWP) between PCWP_New_ and PCWP_Echo_.

**Figure 4 sensors-25-00605-f004:**
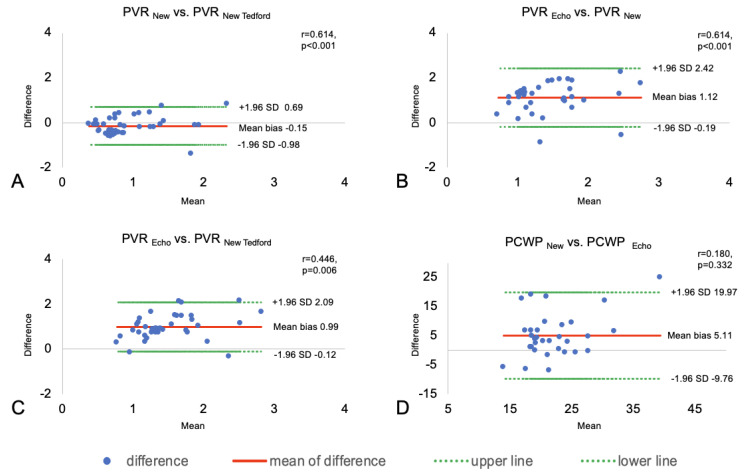
Bland–Altman plot of pulmonary vascular resistance (PVR). (**A**): PVR_New_ and PVR_New Tedford_, (**B**): PVR_New_ and PVR_Echo_, and (**C**): PVR_New Tedford_ and PVR_Echo_. (**D**): Bland–Altman Plot of pulmonary capillary wedge pressure (PCWP) between PCWP_New_ and PCWP_Echo_.

**Table 1 sensors-25-00605-t001:** Characteristics at sensor implantation (clinical characteristics, laboratory parameters, and medication at sensor implantation).

Variables	Study Cohort (n = 13)
**Age,** years	77 ± 4
**Sex**	
- Male	5 (38.5%)
- Female	8 (61.5%)
**BMI,** kg/m^2^	29 ± 5
**NYHA functional class**	
- III	13 (100%)
**Medical history**	
- Previous myocardial infarction	1 (7.7%)
- Previous percutaneous coronary intervention	1 (7.7%)
- Previous coronary artery bypass grafting	1 (7.7%)
- Diabetes mellitus type 2	3 (23.1%)
- Cerebrovascular accident or transient ischemic attack	1 (7.7%)
- Atrial fibrillation	9 (69.2%)
- Arterial hypertension	12 (92.3%)
**Years since heart failure diagnosis**	2 (1.5–5)
**Cause**	
- Ischemic	1 (7.7%)
**Heart rate,** beats per min	72 ± 7
**Systolic blood pressure,** mmHg	129 ± 18
**6-min walking test,** m	307 ± 54
**Type of heart failure**	
- HFpEF	11 (84.6%)
- HFmrEF	2 (15.4%)
- HFrEF	0 (0%)
**Serum creatinine,** mg/dL	1.3 ± 0.5
**eGFR,** mL/min	51 ± 16
**Chronic kidney disease**	
- KDIGO Grade 1	0 (0%)
- KDIGO Grade 2	1 (7.7%)
- KDIGO Grade 3	5 (38.5%)
- KDIGO Grade 4	1 (7.7%)
- KDIGO Grade 5	0 (0%)
**NT-proBNP,** pg/mL	1109 (349–2133)
**Hemotocrit**, %	0.39 ± 0.04
**Hemoglobin**, g/dL	13.0 ± 12.0
**Implanted cardioverter defibrillator**	2 (15.4%)
**Cardiac resynchronization therapy**	2 (15.4%)
**Medical therapy**	
- Beta-blocker	9 (69.2%)
- Renin angiotensin system inhibitor	5 (38.5%)
- Angiotensin converting enzyme inhibitor	4 (30.8%)
- Angiotensin receptor blocker	1 (7.7%)
- Angiotensin-receptor neprilysin inhibitor	4 (30.8%)
- Mineralocorticoid receptor antagonist	8 (61.5%)
- SGLT2 inhibitor	0 (0%)
- Loop diuretic	12 (92.3%)
- Loop diuretic torasemide dose equivalent, mg	20 ± 16
- Thiazide diuretic	3 (23.1%)
- Combined loop and thiazide diuretic	3 (23.1%)

All data are presented in n (%), mean ± standard deviation, or median (IQR). HFpEF, heart failure with preserved ejection fraction. HFmrEF, heart failure with mildly reduced ejection fraction. HFrEF, heart failure with reduced ejection fraction. eGFR, estimated glomerular filtration rate. KDIGO, Kidney Disease Improving Global Outcomes. NT-proBNP, N-terminal fragment of pro-brain natriuretic peptide. SGLT2 inhibitor, sodium glucose-linked transporter-2 inhibitor.

**Table 2 sensors-25-00605-t002:** Hemodynamic and echocardiographic parameters during follow-up.

Variables	At Sensor Implantation	3 Months Prior to SGLT2-I Therapy (Baseline)	Start of SGLT2-I Therapy	3 Months After SGLT2-I Therapy	6 Months After SGLT2-I Therapy	*p* Values
**Echocardiography**						
Quantitative left ventricular ejection fraction, %	60 (53–60)	60 (51–60)	60 (49–60)	60 (50–65)	60 (50–64)	*p* > 0.05 ^(1,2,3,4,5,6)^
TAPSE, mm	22 ± 5	21 ± 5	20 ± 2	21 ± 3	21 ± 5	*p* > 0.05 ^(1,2,3,4,5,6)^
Right ventricular fractional area change, %	41 ± 6	42 ± 5	44 ± 5	38 ± 10	45 ± 6	*p* > 0.05 ^(1,2,3,4,5,6)^
TAPSE/PASP ratio (echo), mm/mm Hg	0.56 ± 0.21	0.60 ± 0.21	0.62 ± 0.24	0.60 ± 0.10	0.58 ± 0.21	*p* > 0.05 ^(1,2,3,4,5,6)^
TAPSE/PASP ratio (CardioMEMS™), mm/mmHg	0.52 ± 0.20	0.48 ± 0.15	0.47 ± 0.16	0.48 ± 0.12	0.48 ± 0.13	*p* > 0.05 ^(1,2,3,4,5,6)^
E/E’mean	12 (10–15)	11.4 (8.5–15.8)	10 (8.3–15.4)	11.0 (10.3–16.0)	9.1 (8.1–15.3)	*p* > 0.05 ^(1,2,3,4,5,6)^
LV-CO, L/min	5.4 (5.1–7.4)	5.2 (4.2–5.7)	4.5 (3.6–5.2)	4.4 (3.8–5.0)	4.6 (4.1–5.3)	*p* > 0.05 ^(1,2,3,4,5,6)^
RV-CO, L/min	6.1 (5.8–7.0)	5.5 (4.5–6.4)	5.1 (4.8–6.8)	4.2 (3.5–5.5)	4.6 (3.45–6.7)	*p* > 0.05 ^(1,2,3,4,5,6)^
LV-SV, mL	87 (78–94)	85 (73–95)	67 (53–77)	75 (58–84)	73 (55–85)	*p* > 0.05 ^(1,2,3,4,5,6)^
RV-SV, mL	91 (80–103)	87 (78–103)	92 (74–117)	82 (62–98)	78 (53–97)	*p* > 0.05 ^(1,2,3,4,5,6)^
Pulmonary arterial capacitance (PAC, Echo-SVRV/CardioMEMS™-Pulse pressure), mL/mmHg	-	3.5 (3.1–5.2)	3.8 (3.4–6.2)	3.5 (2.4–3.7)	3.3 (2.2–4.6)	*p* > 0.05 ^(1,2,3,4,5,6)^
**CardioMEMS™-measured values**						
Systolic PAP, mmHg		46 ± 12	45 ± 14	45 ± 13	45 ± 9	*p* > 0.05 ^(1,2,3,4,5,6)^
Diastolic PAP, mmHg		22 ± 7	22 ± 8	21 ± 7	21 ± 5	*p* > 0.05 ^(1,2,3,4,5,6)^
Mean PAP, mmHg		32 ± 9	32 ± 10	31 ± 10	31 ± 7	*p* > 0.05 ^(1,2,3,4,5,6)^
Pulse pressure, mmHg		24 ± 7	24 ± 7	25 ± 7	24 ± 6	*p* > 0.05 ^(1,2,3,4,5,6)^
Proportional pulse pressure		0.52 ± 0.07	0.52 ± 0.07	0.54 ± 0.07	0.53 ± 0.07	*p* > 0.05 ^(1,2,3,4,5,6)^
**Comparison of different methods**						
PVR_Echo_, WU	1.8 (1.3–2.4)	1.7 (1.5–2.1)	2.1 (1.6–2.5)	1.9 (1.8–2.2)	2.20 (1.18–2.63)	*p* > 0.05 ^(1,2,3,4,5,6)^
PVR_New_, WU		0.7 (0.4–1.1)	0.8 (0.5–0.9)	0.6 (0.4–1.3)	0.76 (0.55–1.53)	*p* > 0.05 ^(1,2,3,4,5,6)^
PVR_New Tedford_, WU		1.0 (0.7–1.1)	0.9 (0.6–1.0)	1.0 (0.9–1.4)	0.95 (0.75–1.55)	*p* > 0.05 ^(1,2,3,4,5,6)^
PCWP_New_		23 (19–25)	23 (19–27)	20 (18–29)	23 (21–27)	*p* > 0.05 ^(1,2,3,4,5,6)^
PCWP_New echo_		14 (10–20)	20 (16–22)	21 (18–23)	17 (16–25)	*p* > 0.05 ^(1,2,3,4,5,6)^

All data are presented in n (%), mean ± standard deviation, or median (IQR). PVR, pulmonary vascular resistance. PCWP, pulmonary capillary wedge pressure. TAPSE, tricuspid annular plane systolic excursion. PASP, systolic pulmonary artery pressure. LV, left ventricular. RV, right ventricular. CO, cardiac output. p^1^ = 3 months prior to vs. day of start of SGLT2 inhibitor; p^2^ = 3 months prior to vs. 3 months after start of SGLT2 inhibitor; p^3^ = 3 months prior to vs. 6 months after start of SGLT2 inhibitor; p^4^ = day of start vs. 3 months after initiation of SGLT2 inhibitor; p^5^ = day of start vs. 6 months after start of SGLT2 inhibitor; p^6^ = 3 months after vs. 6 months after start of SGLT2 inhibitor. ANOVA for repeated measurements and post hoc Bonferroni test or Friedman test for repeated measurements, *p*-values presented with Bonferroni correction for repeated measurements. PVR_Echo_ = TRV/TVIRVOT × 10 + 0.16; PVR_New_ = RC time × PP_New_/SV_New_; PCWP_New_ = PAPd_CardioMEMS_ − DPG_New_; PCWP_Echo_ = E/E’ _Follow-up_ × PCWP_RHC_/E/E’_baseline._

**Table 3 sensors-25-00605-t003:** Laboratory and clinical parameters during follow-up.

Variables	At Sensor Implantation	3 Months Prior toInitiation of SGLT2-I Therapy	Initiation of SGLT2-I Therapy	3 Months After Initiation of SGLT2-I Therapy	6 Months After Initiation of SGLT2-I Therapy	*p* Values
NYHA functional class						
- I		0 (0%)	0 (0%)	0 (0%)	0 (0%)	*p* > 0.05 ^(1,2,3,4,5,6)^
- I–II		0 (0%)	1 (7.7%)	1 (7.7%)	0 (0%)	
- II		1 (7.7%)	2 (15.4%)	3 (23.1%)	4 (30.8%)	
- II–III		2 (15.4%)	3 (23.1%)	2 (15.4%)	2 (15.4%)	
- III	13 (100%)	10 (76.9%)	6 (46.2%)	7 (53.8%)	7 (53.8%)	
- IV		0 (0%)	1 (7.7%)	0 (0%)	0 (0%)	
NT-proBNP, pg/mL	1109 (349–2133)	1412 (562–2011)	1536 (790–3152)	1170 (754–2373)	1017 (568–3513)	*p* > 0.05 ^(1,2,3,4,5,6)^
Hematocrit, %	0.39 ± 0.04	0.38 ± 0.02	0.39 ± 0.03	0.38 ± 0.03	0.40 ± 0.04	*p* > 0.05 ^(1,2,3,4,5,6)^
Hemoglobin, g/dL	13.0 ± 12.0	12.8 ± 0.7	12.8 ± 1.1	13.2 ± 1.3	13.1 ± 1.3	*p* > 0.05 ^(1,2,3,4,5,6)^

All data are presented in n (%), mean ± standard deviation, or median (IQR). NYHA, New York Heart Association. NT-proBNP, N-terminal pro-B natriuretic peptide. p^1^ = 3 months prior to vs. day of start of SGLT2-I; p^2^ = 3 months prior to vs. 3 months after start of SGLT2-I; p^3^ = 3 months prior to vs. 6 months after start of SGLT2-I; p^4^ = day of start vs. 3 months after initiation of SGLT2-I; p^5^ = day of initiation vs. 6 months after start of SGLT2-I; p^6^ = 3 months after vs. 6 months after initiation of SGLT2-I. ANOVA for repeated measurements and post hoc Bonferroni test or Friedman test for repeated measurements, *p*-values presented with Bonferroni correction for repeated measurements. Student’s *t*-test for non-repeated measurements.

## Data Availability

The data of this study are combined together and analyzed within the center registry (NCT03020043) and can be obtained by written request.

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
