# Peer review of "Case Series Evaluating the Relationship of SGLT2 Inhibition to Pulmonary Artery Pressure and Non-Invasive Cardiopulmonary Parameters in HFpEF/HFmrEF Patients—A Pilot Study"

_sensors, 2025, doi:10.3390/s25030605_

Round 1
Reviewer 1 Report
Comments and Suggestions for Authors
The manuscript entitled “Case series evaluating the relationship of SGLT2 inhibition to pulmonary artery pressure and non-invasive cardiopulmonary parameters in HFpEF/HFmrEF patients” was evaluated. The article presents relevant information; however, some adjustments are necessary for it to be published.
Below are some suggestions and issues to be adjusted in the article by the authors:
Abstract
1- In line 18 of the summary, before presenting the abbreviation “SGLT2”, write the full name: Sodium–glucose cotransporter 2.
2- Also check other acronyms and write the full name first, such as NYHA, HFpEF and HFmrEF.
3- In the abstract results, present the “p” and “r” values of the tests performed.
Introduction
4- Write the acronyms “HFpEF and HFmrEF” in the introduction before citing the abbreviation.
Materials and Methods
5- The inclusion and exclusion criteria for study participants were not described between lines 72 and 77. Please include them.
6- The sample size calculation and the number of people selected to participate in the research were also not found in the methodology.
7- Include the link to the clinical trial registry between lines 140 and 143.
8- Adjust the formatting of the description in the statistical analyses. The second paragraph is not configured correctly.
9- Include the study flowchart.
Results
10- Regarding the tables, please remove the lines between the variables. Leave lines only at the beginning and end of the table. In table 1, write the word “variables” at the top.
11- In table 2, put the variables in text: Right heart catheter (at sensor implantation). This information is making the table very cluttered and since there is no statistical analysis involving these variables, there is no need to leave it on the table. Also in this table, since all the “p” values are all above 0.05, I believe you can present the value only once in front of the items and present it as follows: p>0.05(1,2,3,4,5,6).
12- In Figure 1, since no statistical differences were found, I believe it can be removed from the article and the information about it can be included in the text.
13- In Table 3, follow the same recommendations presented for Table 2.
14- In Figure 3, add the “p” and “r” values to each figure to make it easier for the reader. Also remove the grid lines in the background of the image, this makes the figure cleaner. In the 3D figure, the X axis could reach up to 40 to improve the amplitude of the graphical presentation.
15- In Figure 4, I did not find the information about the comparisons, please include in the text what readers should find in the image and insert the “p” values of the comparisons in the image.
Discussion
16- Regarding limitations, I believe that if this number of people is within the minimum sample size calculation, it would not be a limitation, so it is important to include this information in the methodology. In addition, other information could also be included as limitations if the inclusion and exclusion criteria are clear.
17- Please review the study to adjust the formatting according to the journal's standards. There are several places where the formatting is presented in different ways.
Reviewer 2 Report
Comments and Suggestions for Authors
This manuscript "Case Series Evaluating the Relationship of SGLT2 Inhibition to Pulmonary Artery Pressure and Non-Invasive Cardiopulmonary Parameters in HFpEF/HFmrEF Patients" contains several significant issues that require careful revision.
Firstly, the study concludes that no significant changes were observed in key cardiopulmonary parameters following SGLT2-I initiation. The absence of meaningful results diminishes its impact and contribution.
Although negative findings are valuable, the manuscript fails to convincingly articulate their relevance or implications for clinical practice or future research.
The cohort size (13 patients) is too small to draw reliable or generalizable conclusions. This limitation undermines the study’s statistical power and validity.
The choice of a small cohort and reliance on a limited timeline (9 months) raise concerns about whether the study was adequately designed to detect significant hemodynamic changes.
The study compares three methods for calculating PVR and PCWP but does not justify the need for these comparisons or their relevance to the research objectives.
The use of Bland-Altman plots and Spearman’s correlation for evaluating methods is described but not adequately linked to the study’s overall aim, making the methodology appear disjointed.
The manuscript uses technical terms like "RV-PA coupling" and "PAC" without providing sufficient context or explanation, limiting accessibility for a broader audience.
The age of the cohort is incorrectly reported as "mean age 774 years," suggesting a lack of attention to detail in manuscript preparation.
This research does not convincingly explain how the findings (or lack thereof) can influence clinical practice or contribute to our understanding of SGLT2 inhibitors in heart failure management.
The authors fail to explore alternative explanations for lack of observed changes, leaving the reader with unanswered questions.
The conclusions section provides little insight beyond stating that no changes were observed. It does not discuss potential reasons for these results or propose meaningful directions for future research.
The recommendation for larger cohorts and repeated right heart catheterization, while valid, is generic and does not offer specific strategies or hypotheses to guide further studies.
The glaring error in reporting the cohort's mean age reflects poor proofreading and quality control, undermining the manuscript's credibility.
The abstract does not clearly outline the primary hypothesis, making it difficult to evaluate whether the study achieved its objectives.
Comments on the Quality of English LanguageThe English could be improved to more clearly express the research.
Round 2
Reviewer 1 Report
Comments and Suggestions for Authors
After checking the adjustments made to the manuscript by the authors, my suggestion is that the article be accepted.
Author Response
Comment: After checking the adjustments made to the manuscript by the authors, my suggestion is that the article be accepted.
Response: Thank you for your thoughtful review and for taking the time to assess the revisions made to our manuscript. We greatly appreciate your positive feedback and are pleased to hear that you are satisfied with the adjustments. Your support and constructive suggestions have been invaluable in improving the quality of the article. We are grateful for your assistance throughout the review process.
Reviewer 2 Report
Comments and Suggestions for Authors
All my comments and suggestions have been adequately addressed. Kindly double-check the Figures and Tables.
Author Response
Comment: All my comments and suggestions have been adequately addressed. Kindly double-check the Figures and Tables.
Response: Thank you for your thoughtful feedback and for taking the time to review the revised manuscript. We are glad to hear that the majority of your comments and suggestions have been adequately addressed.
In response to your note regarding the Figures and Tables, we have thoroughly double-checked them and made sure that all are accurate and correctly formatted. We appreciate your attention to detail and believe that these revisions further improve the manuscript. Thank you once again for your valuable input.